# T_R3-56_ and Treg Regulatory T Cell Subsets as Potential Indicators of Graft Tolerance Control in Kidney Transplant Recipients

**DOI:** 10.3390/ijms251910610

**Published:** 2024-10-02

**Authors:** Valentina Rubino, Flavia Carriero, Anna Teresa Palatucci, Angela Giovazzino, Fabrizio Salemi, Rosa Carrano, Massimo Sabbatini, Giuseppina Ruggiero, Giuseppe Terrazzano

**Affiliations:** 1Dipartimento di Scienze Mediche Traslazionali, Università di Napoli “Federico II”, 80131 Napoli, Italy; valentina.rubino@unina.it (V.R.); angela.giovazzino@unina.it (A.G.); 2Dipartimento di Scienze Della Salute, Università Della Basilicata, 85100 Potenza, Italy; flavia.carriero@unibas.it (F.C.); anna.palatucci@unibas.it (A.T.P.); giuseppe.terrazzano@unibas.it (G.T.); 3Percorso Clinico Assistenziale in Nefrologia e Trapianto Renale, Azienda Ospedaliera Universitaria “Federico II”, 80131 Napoli, Italy; fabrizio.salemi@unina.it (F.S.); carrano.rosa5@unina.it (R.C.); 4Dipartimento di Sanità Pubblica, Sezione di Nefrologia, Università di Napoli “Federico II”, 80131 Napoli, Italy; sabbatin@unina.it

**Keywords:** regulatory T cells, Treg, T_R3-56_, kidney transplant recipients, immune profile

## Abstract

Identification of early signatures of immune rejection represents a key challenge in the clinical management of kidney transplant. To address such an issue, we enrolled 53 kidney transplant recipients without signs of graft rejection, no infectious episodes and no change in the immunosuppressive regimen in the last 6 months. An extensive immune profile revealed increased activation of the T cells, a decreased amount and growth ability of the Treg and a higher level of the T_R3-56_ regulatory T cell subset, described by us as involved in the preferential control of cytotoxic T lymphocytes. In renal transplant recipients, the high level of the T_R3-56_ cells associates with a reduction in both the amount and the growth ability of the Treg. Moreover, when the transplanted subjects were categorised according to their stable or unstable disease status, as defined by changes in serum creatinine ≥0.2 mg/dL in two consecutive detections, a higher T_R3-56_ level and defective Treg growth ability were observed to characterise patients with unstable graft control. Further studies are required to substantiate the hypothesis that immune profiling, including T_R3-56_ evaluation, might represent a valuable diagnostic tool to identify patients at risk of developing significant anti-donor allo-immune responses.

## 1. Introduction

Immune-mediated processes have been largely observed to underlie kidney homeostasis maintenance, also playing a key role in the progression of chronic kidney disorders. Indeed, the derangement of the complex balance involving regulatory cell subsets and the activity of adaptive immune effectors have been considered critical for the pathogenesis of kidney diseases [1]. 

Kidney transplantation represents the main therapeutic option to control end-stage kidney disease [2]. In this context, recognition of allo-specificities by recipient immune effectors, despite the progress in immune-modulating approaches, remains a leading cause of allograft injury and loss [3]. Although the antibody-mediated (humoral) rejection is considered the leading cause of progressive graft dysfunction and loss [4,5], the involvement of T cell-dependent immunity in both early and late events contributing to graft rejection remains a critical issue [4]. Indeed, the key role of helper T cells in the activation of an effective humoral response has been largely established [6,7]. In this context, the availability of valuable criteria to identify in time early immune-mediated injuries in kidney transplant recipients represents a still unmet target. 

Fine-tuning of the immune response is usually obtained by multiple regulatory processes, all belonging to the immune tolerance network [8,9,10] whose perturbation is expected to associate with immune-mediated tissue injury. In this context, the key role of regulatory populations in the prevention of immune-dependent damaging events has been largely shown [11]. 

Regulatory cells represent a heterogeneous group of differentiated T cell subsets including the CD25^+^CD4^+^ regulatory T (Treg) cells, constitutively expressing the forkhead box P3 (Foxp3) transcription factor [12]. This cell subset has been largely demonstrated to control the immune-effector response in terms of clonal expansion, differentiation, cytokine profile and tissue migration and is indispensable for the maintenance of immune self-tolerance [13]. Recently [14,15], we described how human T cells co-expressing CD3 and CD56 molecules represent a novel regulatory subset, the T_R3-56_, able to preferentially modulate cytotoxic function and cytokine production by cytotoxic T cells (CTLs). The involvement of T_R3-56_ in the pathogenesis of Type 1 diabetes [14] as well as in immune-mediated haematological disorders has also been described by us [16,17] and by others [18].

CTLs play a relevant role in mediating transplant damage [19,20,21]. In this context, the expression of activation molecules like CD25, CD69, CD154, CD95 on the surface of circulating CTLs from kidney recipients, the increased frequency of terminally differentiated memory CD8 T cells [20] and a deranged Treg/CTL ratio in the transplanted tissue [22,23] have already been proposed as relevant markers to identify early immune-mediated injuries in kidney transplant recipients. Since the T_R3-56_ regulatory T cells have been observed to preferentially control CTL functions [14,15,16,17], the possibility that these cells may be involved in the control of immune-mediated damaging processes of kidney grafts needs to be explored. 

Therefore, we analysed the immune profile of 53 kidney transplant recipients without signs of graft rejection, with no infectious episodes and no change in the immunosuppressive regimen in the last 6 months. Our analysis, in order to investigate the immune profile features, likely associated with early immune-mediated injuries of the transplanted kidney, focused on circulating immune regulatory T cell subsets, as represented by the Treg and the T_R3-56_ populations, as well as the adaptive and innate immune effectors. 

## 2. Results

### 2.1. Higher Amount of Circulating T_R3-56_ Regulatory T Cells and Decreased Level of the Treg Lymphocytes Characterise Kidney Transplanted Subjects Showing No Signs of Graft Rejection

Figure 1 and Appendix A depict the comprehensive immune profile analysis performed in the cohort of 53 kidney transplant recipients showing no signs of graft rejection, no infectious episodes and no change in the immunosuppressive regimen in the last 6 months, in comparison with 20 age/sex matched healthy controls. 

As shown in Figure 1A–C, comparative analysis with controls showed that CD4^+^ T cells were significantly decreased in transplant recipients both as a percentage (67.04 ± 1.74 in controls vs. 52.17 ± 2 in the kidney recipients; *p* < 0.0001), and as an absolute number (977 ± 30.52 10^9^/L in controls vs. 765 ± 59.11 10^9^/L in the kidney recipients; *p* < 0.0001). Concurrently, there was an increase in the circulating CTL percentage (33.47 ± 0.88 vs. 47.83 ± 2 in the kidney recipients; *p* < 0.0001) and number (486.5 ± 18.19 10^9^/L vs. 678.3 ± 60.96 10^9^/L; *p* < 0.0001), and a decrease in the CD4/CD8 ratio (2.16 ± 0.15 in controls vs. 1.23 ± 0.09 in the kidney recipients; *p* < 0.0001).

Analysis of the surface expression of the CD54 molecule (Figure 1D), largely associated with antigen-dependent triggering of the T cells [25,26], revealed significant activation of both, CD4^+^ (2.51 ± 0.75 in controls vs. 7.44 ± 0.37 in the kidney recipients; *p* < 0.0001) and CTL (6.72 ± 1.39 in controls vs. 13.77 ± 0.76 in the kidney recipients; *p* < 0.0001). We then focused on the regulatory subsets, as represented by the Treg as well as by the T_R3-56_ T cell population. In comparison with healthy subjects (Figure 1E,F), the T_R3-56_ regulatory T cell subset was significantly increased in the transplant recipients, both as a percentage (8.32 ± 0.89 in controls vs. 13.40 ± 1.43 in the kidney recipients; *p* < 0.005) and as a number (94.05 ± 10.61 10^9^/L in controls vs. 185.6 ± 23.34 10^9^/L in the kidney recipients; *p* < 0.005) whereas Treg lymphocytes (Figure 1G,H) were reduced in transplanted subjects either as a percentage (7.66 ± 0.53 in controls vs. 5.05 ± 0.42 in the kidney recipients; *p* < 0.0001) or as a number (68.37 ± 5.24 10^9^/L in controls vs. 42.13 ± 5.54 10^9^/L in the kidney recipients; *p* < 0.0001).

A key feature of the Treg subset is its high growth ability [27,28]. Thus, we evaluated this parameter in both circulating regulatory T cell subsets (Treg and T_R3-56_), by detecting their ki67 intracellular expression. As shown, we observed (Figure 1J) a significant reduction in the ki67 intracellular expression in the circulating Treg lymphocytes of transplant recipients (11.23 ± 0.89 in controls vs. 8.73 ± 0.77 in the kidney recipients; *p* < 0.05). However, (Figure 1I) no significant difference (4.17 ± 0.54 in controls vs. 3.93 ± 0.51 in the kidney recipients) was revealed in the growth ability of the T_R3-56_ subset in comparison with healthy controls. 

As shown in the Appendix A, significantly decreased levels of the B lymphocytes (Appendix A), a lower percentage of circulating iNKT cells (Appendix A), no difference in the number of iNKT (Appendix A), as well as in the percentage and amount of the circulating NK effectors (Appendix A) were observed in the transplant recipients, in comparison with controls. Moreover, the comparative analysis of the growth ability of innate and adaptive lymphocytes between kidney recipients and controls revealed no significant difference in ki67 intracellular expression by CD4^+^ and CTL T cells (Appendix A), or in B lymphocytes (Appendix A) or NK effectors (Appendix A). 

Thus, as compared with healthy controls, the immune profile of the cohort of kidney transplant recipients revealed the presence of activated adaptive effectors, of an increased amount of the T_R3-56_ lymphocytes and a decreased level and growth ability of the Treg subpopulation; in addition, increasing CTL, a reduced level of CD4^+^ T cells, of B cells and of iNKT lymphocytes, was observed. 

### 2.2. Decreased Growth Ability of the Treg Subset Characterises the Subgroup of Transplanted Subjects with a Higher Amount of the Circulating T_R3-56_ Regulatory T Cells

The cohort of kidney transplant recipients was characterised by increased activation of the CTL, higher levels of circulating T_R3-56_ regulatory T cells and a decreased amount and growth ability of the Treg population. Accordingly, we focused our investigation on the immune profile associated, in our cohort, with the highest levels of the circulating T_R3-56_ lymphocytes. In this context, kidney transplant recipients were categorised according to their level of circulating T_R3-56_ in two sub-groups characterised by a T_R3-56_/T cell ratio higher or similar to the controls. The cut-off value (9.16% of the T lymphocytes) was arbitrarily established, as detailed in the Patients and Methods Section, by increasing by three SEM the median value observed in the healthy individuals enrolled in the study.

Light and dark grey columns show, in Figure 2, the results obtained in the group of transplant recipients characterised by T_R3-56_ lymphocyte levels similar to (≤9.16% of the T lymphocytes) or higher (>9.16% of the T lymphocytes) than the controls, respectively. Figure 2A,B show that kidney transplant recipients, regardless of the amount of circulating T_R3-56_ regulatory T cells, showed a decreased amount of the CD4^+^ T cells, increasing levels of the CTL, a reduced CD4/CD8 ratio (Figure 2C) and a lower percentage and number of the Treg population (Figure 2E,F). Moreover, as depicted in Figure 2D, increasing activation of the T cell effectors, as evaluated by their CD54 expression, was found in the patients, as compared with the controls; however, a more consistent increase in CD54 expression was observed in the CD4^+^ T lymphocytes of the transplant recipients with a higher T_R3-56_ level (6.39 ± 0.61 in the kidney recipients with a T_R3-56_ level ≤ 9.16% of the T lymphocytes vs. 8.13 ± 0.43 in the kidney recipients with T_R3-56_ > 9.16% of the T lymphocytes; *p* < 0.05). 

As shown in Appendix A, the significant decrease in circulating B cells, found in the kidney transplant recipients versus controls, has been found to be more consistent in the subgroup of transplant recipients characterised by higher circulating T_R3-56_ regulatory T cells (6.70 ± 0.95 in the kidney recipients with a T_R3-56_ level ≤9.16% of the T lymphocytes vs. 4.30 ± 0.56 in the kidney recipients with the highest T_R3-56_ level; *p* < 0.05). Conversely, no difference was found in the level of iNKT (Appendix A) and of NK lymphocytes (Appendix A), in the patients, as compared with the controls, regardless of their level of circulating T_R3-56_ lymphocytes. Similarly, (Appendix A), no difference was revealed in the growth ability of CD4^+^, CTL, iNKT and B lymphocytes between controls and patients, independently of their level of circulating T_R3-56_ lymphocytes. 

Notably (Figure 2G), only the kidney transplant recipients showing a higher T_R3-56_ level (>9.16% of the T lymphocytes) revealed a significant decreased growth ability of the Treg subset (11.08 ± 1.50 in the kidney recipients with a T_R3-56_ level ≤ 9.16% of the T lymphocytes vs. 7.34 ± 0.75 in the kidney recipients with the highest T_R3-56_ level; *p* < 0.05). 

Thus, in renal transplant recipients, higher T_R3-56_ levels are preferentially associated with increased activation of the CD4^+^ T cells and decreased growth ability of the Treg subset. 

### 2.3. The Presence of Higher T_R3-56_ Levels in Kidney Transplanted Subjects Showing No Signs of Graft Rejection Associates with Early Signs of Unstable Graft Tolerance

Our immune profile analysis showed that, in our cohort of kidney transplant recipients, a higher amount of the circulating T_R3-56_ regulatory T cells preferentially associates with a decreased number and growth ability of the Treg, as well as with increased activation of the CD4^+^ T lymphocytes. Thus, we investigated the possibility that the level of circulating T_R3-56_ might represent a valuable criterion to identify kidney transplant recipients showing early signs of unstable control of the graft. With this aim, the kidney recipients were classified, according to their laboratory and clinical data, into a Stable and Unstable group, as detailed in the Patient and Method Section. Briefly, the Stable group was represented by subjects with a stable renal function and urinary parameters, while the Unstable group included patients showing changes in the serum creatinine level ≥0.2 mg/dL and/or proteinuria > 100 mg/day in 24-h urinary samples in two consecutive evaluations, despite no clinical predisposing conditions. 

As shown in Figure 3, comparative analysis of the immune profile of the kidney recipients, grouped according to their Stable versus Unstable clinical conditions, revealed a significant difference between the two patient subgroups in the CTL number (546.89 ± 38.84 × 10^9^/L in the kidney recipients with a Stable disease vs. 713.8 ± 80.43 × 10^9^/L in the kidney recipients with an Unstable disease; *p* < 0.05), higher in the individuals belonging to the Unstable disease group, also showing (Figure 3B,C), a lower CD4/CD8 ratio (1.45 ± 0.14 in the kidney recipients with a Stable disease vs. 0.99 ± 0.09 in the counterparts with an Unstable disease; *p* < 0.05).

As shown (Figure 3E,F), a significant increase in the circulating T_R3-56_ cells as a percentage (9.75 ± 1.31 in the kidney recipients with the Stable disease vs. 17.04 ± 2.36 in the counterparts with an Unstable disease; *p* < 0.05) as well as number (120.6 ± 19.27 10^9^/L in the kidney recipients with Stable disease vs. 254.8 ± 40.32 10^9^/L in the subgroup with Unstable disease; *p* < 0.01), associated with (Figure 3J), significant reduced Treg growth ability (10.77 ± 1.14 in the kidney recipients with Stable disease vs. 6.29 ± 0.78 in the kidney recipients with Unstable disease; *p* < 0.005), characterise the patient subgroup with Unstable disease. No difference has been observed in the amount of B cells, iNKT and NK lymphocytes (Appendix A) as well as in the growth ability of adaptive and innate immune effectors (Appendix A) comparing the subgroup of transplanted subjects with a Stable disease with the counterparts with an Unstable disease. 

To investigate whether a high level of the circulating T_R3-56_ cells, as defined by increasing by three SEM the median value obtained in healthy controls, might be significantly associated with an unstable graft control, we analysed the presence of circulating T_R3-56_ cells at a percentage > 9.16% of T lymphocytes in kidney transplant recipients categorised in the Unstable versus the Stable disease subgroup, as defined in the Patients and Methods Section. As shown (Table 1), 19 out of the 25 subjects with an Unstable disease showed a T_R3-56_ cell percentage > 9.16% of circulating T lymphocytes (defined higher T_R3-56_ level), as compared with 10 out of the 25 subjects with a Stable renal disease (*p* < 0.05 by Fisher exact test; Odd Ratio 4.75; 95% CI 1.384 to 14.37).

Such data suggest that the level of circulating T_R3-56_ cells might represent a valuable criterion to identify kidney transplant recipients with early signs of immune mediated damages to the graft, in the absence of any clinical and/or laboratory rejection treat.

## 3. Discussion

This study reveals that, in a cohort of 53 kidney transplant recipients not showing any clear clinical/laboratory sign of kidney rejection, higher levels of circulating T_R3-56_ regulatory T cells are associated with unstable control of the transplanted kidney. This instability condition was unveiled by changes in the serum creatinine level ≥0.2 mg/dL and/or in proteinuria >100 mg/day in 24-h urinary samples in two consecutive bi-monthly evaluations, despite no clinical predisposing conditions. These findings propose that the evaluation of the circulating T_R3-56_ T cell subset might serve as a potential indicator of early immune-mediated processes that could potentially impact graft tolerance. 

Kidney transplant represents a major therapeutic option to effectively treat renal end-stage disease [1,2]. In this context, despite advancements in immune-modulating approaches, immune-mediated damaging processes continue to pose a significant challenge for allograft injury and loss. Moreover, valuable criteria to identify in time early immune-mediated injuries in kidney transplant recipients are still lacking. 

The key role of T lymphocytes, able to orchestrate the whole immune response by controlling both humoral and cytotoxic activities, has been largely demonstrated [6,7]. 

T cell-dependent processes have been described as depending on multiple immune-regulatory networks involving different regulatory T cell subsets [10,11,12,13], only in part characterised by the expression of the Foxp3 transcription factor [15,29,30,31,32,33]. Moreover, the role of multiple inflammatory pathways, early after transplantation [34] and in long-term transplant recipients [35,36,37,38], has been largely referred to.

Co-expression of CD3 and CD56 molecules has been revealed by us to define the T_R3-56_ regulatory T cell subset, preferentially involved in the control of cytotoxic T cell effectors in autoimmunity [14] as well as in haematologic disorders [16,17,18]. This study investigates the T_R3-56_ subpopulation in a cohort of 53 kidney allograft recipients showing no signs of graft rejection, no infectious episodes and no change in their immunosuppressive regimen in the last 6 months. 

Our extensive comparative immune profile of the kidney transplant recipients versus healthy controls revealed several peculiar features: i. increased expression of the CD54 molecule, largely associated with antigen-dependent T cell activation, by T cell effectors [25,26]; ii. increasing number and percentage of cytotoxic T cells; iii. decreased amount and growth ability of the Treg; iv. increasing percentage and number of the T_R3-56_ regulatory T cell subset. 

The constant availability of allo-antigens, as represented by the graft, has to be considered as a key feature underlying the immune scenario observed by us. Indeed, a low-grade pro-inflammatory microenvironment characterises the transplanted organ even in the presence of an effective immune-modulation therapy [1,2]. A significant association of CD4^+^ and CTL activation with a reduction of the Treg subset has been largely found to underlie long-standing infection, transplantation and autoimmune diseases in human [34,35,36,37,38,39] and animal models [40,41]. In this context, the increasing T_R3-56_ level in the transplanted subjects, when considering the immune-modulating role of such a T cell subset and the concomitant increase in the CTL population, might underlie an attempt to restore/maintain graft immune tolerance control in the presence of defective Treg-mediated immune-suppression. 

Compelling evidence indicates growth ability as a major feature of the Treg population [38,39]. Such a treat has been largely associated with the need for a dynamic regulation of this T cell subset, specifically involved in the maintenance of a complex homeostatic balance. In this context, the decreased growth ability of the Treg subset might be likely related to an early derangement of the immune tolerance control of the graft in kidney recipients. Accordingly, the presence of an increasing level of activated helper T cells, the ones responsible for immune orchestration [6,7] in subjects showing highest amount of the T_R3-56_ T cell regulatory population, might represent an attempt, in subjects showing early signs of graft functional recognition, to restore transplant tolerance control, thus avoiding clinical rejection episodes. 

We found that categorising kidney transplant recipients based on their stable control of the graft, evaluated by changes in creatinine levels and 24 h proteinuria over two consecutive bi-monthly analyses, revealed a significant association of a higher T_R3-56_ amount with an increasing CTL level and a decreased growth ability of the Treg population. Furthermore, our study revealed a significant association of the highest T_R3-56_ levels with unstable graft control. 

These observations, as a whole, propose a scenario in which early derangement of graft tolerance, in the presence of maintained kidney functional effectiveness, might be associated with increasing levels of the T_R3-56_ regulatory T cell population and a low growth ability of the Treg subset. In this context, to minimise the pro-inflammatory effect related to dialysis pre-transplant treatment, a transplant vintage exceeding one year was considered in the enrolment criteria. Moreover, the absence of autoimmune diseases, as well as of viral infections in the enrolled cohort, was expected to focus the investigation on the peculiar immune traits/cell subsets underlying the early events associated with deranged transplant tolerance. The mechanisms underlying such a complex scenario need further investigation. 

The concomitant immune-modulating therapy that characterises the cohort of transplant recipients analysed by us is probably related to an inability to observe an inverse association of the T_R3-56_ amount with the activation level of their CTL target, as demonstrated in the haematological model [16,17].

The observation that in kidney transplant recipients, high levels of circulating T_R3-56_ regulatory T cells significantly associate with a decreased Treg growth ability, proposes T_R3-56_ evaluation as a potential marker of early defective graft tolerance control. Larger evaluations are needed in order to propose the potential employment of the analysis of the circulating T_R3-56_ T cell regulatory subset as a valuable indicator of graft stable control in kidney transplant recipients. 

## 4. Materials and Methods

### 4.1. Patients

The study was carried out on 53 renal transplant recipients, all first transplanted from cadaver donors, in a regular follow-up at the Percorso Clinico Assistenziale in Nefrologia e Trapianto Renale, Azienda Ospedaliera Universitaria “Federico II”, c/o the Dipartimento di Salute Pubblica of the Università Federico II (Napoli, Italy). Inclusion criteria were: age 18–65 years; transplant vintage: >1 year, with a bi-monthly clinical and laboratory control in the last 6 months; plasma creatinine < 3 mg/dL; haemoglobin value > 11 g/dL; white cell count > 4000/μL (neutrophils > 2000/μL); platelet count > 75.000/μL; absence of clinical signs of graft rejection, of infectious episodes and no change in the immunosuppressive regimen in the last 6 months. The subjects were randomly enrolled regardless of their dialysis vintage (usually between 2 and 4 years) as well as the pathogenic condition underlying the end-stage kidney failure requiring a kidney transplant. No anti-thymocyte serum was employed in the induction treatment. Exclusion criteria were: previous or combined transplantation; Panel Reactive Antibodies (PRAs) > 25% and/or presence of Donor Specific Antibodies (DSAs) at transplantation; presence of proteinuria exceeding 300 mg/day on 24-h samples; presence of hyperlipidaemia (baseline cholesterol and/or triglycerides values exceeding 220 and 200 mg/dL, respectively); evidence of autoimmune diseases or viral infections. 

On the basis of these criteria, 53 consecutive patients, in a regular follow-up, were included in the study. These subjects were subsequently divided into two groups according to their laboratory data: patients with stable renal function and urinary parameters (Stable Group), and patients showing changes ≥ 0.2 mg/dL in serum creatinine level and/or > 100 mg/day in proteinuria 24-h urinary samples in two consecutive evaluations, despite no clinical predisposing condition (worsening hypertension, recurrence of the underlying renal disease, cardiovascular disease). These patients represented the Unstable Group. Plasma creatinine concentration was evaluated by an autoanalyser with a modified Jaffè method, urinary protein excretion by PCR ([urinary protein/urinary creatinine] × 1000, mg/mM) method; glomerular filtration rate was estimated with EPI-CKD formula. Immune-modulating treatments included Corticosteroids and Calcineurin inhibitors, as detailed in Table 2. Demographic and laboratory data of the enrolled subjects are shown in Table 2. The study, conducted in agreement with Good Clinical Practice guidelines, was approved by the Ethics Committee of the University Federico II of Naples (Protocol number: 66/11). All the procedures were in accordance with the Declaration of Helsinki, as revised in 2008. A total of 20 blood donors, age- and sex-matched with the patients, were enrolled into the study, as healthy controls; their baseline cholesterol and triglycerides values were always <200 mg/dL. All the patients and controls signed their informed consent to the study. 

### 4.2. Cells, Immunofluorescence and Flow Cytometry Analysis

Blood samples were analysed by immunofluorescence by using FITC or Pe-Cy5 anti-human CD3 (BD Pharmingen, clone UCHT1), FITC anti-human CD4 (BD Pharmingen, clone RPA-T4), Pe-Cy7 anti-human CD8 (BD Pharmingen, clone RPA-T8), Pe-Cy5 or Pe-Cy7 anti-human CD56 (BD Biosciences, clone NCAM16.2), PE anti-human CD25 (BD, clone M-A251), PE anti-human CD54 (BD, clone HA58), FITC anti-human Ki-67 (BD, clone B56), all from Becton Dickinson Italia S.p.A., Milano, Italy. PE anti-human Vα24 (Beckman Coulter, clone C15), from Beckman Coulter S.p.a., Milano, Italy; FITC anti-human CD19 (eBioscience, clone HIB19), FoxP3-all (eBioscience, clone PCH101. from Thermo Fisher Scientific Inc., MA, USA. For intracellular detection of FoxP3-all and of Ki-67, a fixation and permeabilization FoxP3 buffer kit (eBioscence, from Thermo Fisher Scientific Inc., MA, USA), was employed according to the manufacturer’s instructions. For the analysis of the CD54 expression level in T lymphocytes, fluorescence data were expressed as a ratio of the mean intensity fluorescence (MIF) value for the CD4 or CD8 T cell population and the control MIF value obtained after staining of the same cell subset with the isotype control mAb, as described [24]. All phenotypes referred to flow cytometry analysis of the lymphocyte population gated using forward (FSC) and side-scatter (SSC) parameters. Flow cytometry evaluation was performed by using an ATTUNE NxT acoustic focusing cytometer (Life Technologies; Thermo Fisher Scientific Inc., MA, USA). Data analysis was performed by using FlowJo Software (V10, LLC). PBMCs, for FoxP3 and ki67 detection, were isolated by centrifugation of the peripheral blood on a Ficoll-Paque cushion (GE Healthcare, Uppsala, Sweden) gradient. This evaluation strategy is expected to allow the direct analysis of the biological complexity of the immune profile in the peripheral blood. Accordingly, we focused on adaptive immune effectors, chiefly responsible for allo-antigen recognition/damaging, by evaluating their surface expression of the CD54 molecule, consistently associated with antigen-dependent T cell activation [25,26].

To evaluate possible oscillations in the results, two independent samples, obtained for each subject, were analysed at a one-week interval and produced substantially comparable results. 

### 4.3. Statistical Analysis

Statistical evaluation of data, by using GraphPad Prism 6.0 software (GraphPad Software, Inc., La Jolla, CA, USA), was performed by a Mann–Whitney test and a Fisher’s two-tailed exact test, as indicated. Two-sided *p* values of less than 0.05 were considered to indicate statistical significance. 

## 5. Conclusions

Recognition of allo-specificities by recipient immune effectors has been largely recognised to underlie allograft injury and loss. Moreover, the availability of valuable criteria to identify in time early immune-mediated injuries in kidney transplant recipients, represents a still unmet target. Here, we show that increased activation of T cells, a decreased amount and growth ability of the Treg and higher level of the T_R3-56_ regulatory T cell subset, consistently associated by us with the preferential control of cytotoxic T lymphocytes, characterise a transplant recipient cohort without signs of graft rejection, no infectious episodes and no change in the immunosuppressive regimen in the last 6 months. In addition, the highest level of the circulating T_R3-56_ regulatory subset specifically associates with a reduction in the amount and growth ability of the Treg. Accordingly, unstable graft control, as defined by changes in serum creatinine ≥ 0.2 mg/dL in two consecutive bi-monthly detections, has been consistently associated with a higher T_R3-56_ level and defective Treg growth ability. Further studies are required to validate the hypothesis that immune profiling, including T_R3-56_ evaluation, might represent an early diagnostic tool to identify patients at risk of developing significant anti-donor allo-immune responses. 

## 6. Limitations

One limitation of the study is represented by the lack of details about the underlying co-morbidities in the graft recipient cohort. Further studies are needed to deeply investigate such an issue. 

## Figures and Tables

**Figure 1 ijms-25-10610-f001:**
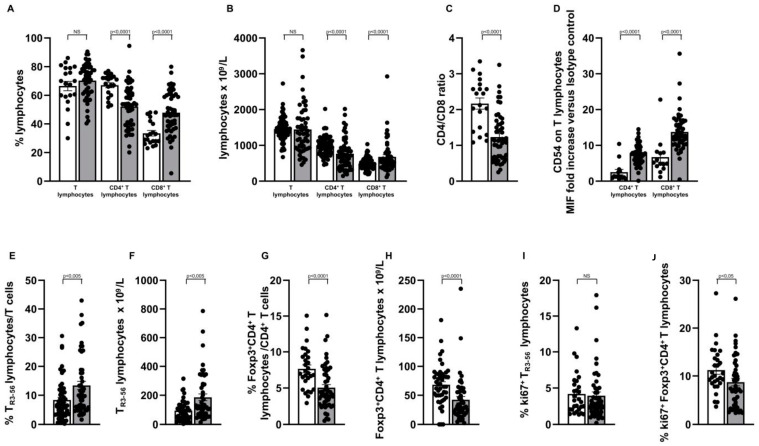
Increased amount of the circulating CTL and of the T_R3-56_ regulatory T cells, higher expression of the CD54 activation molecule on T cell effectors and decreased amount and growth ability of the Treg subset characterise a cohort of allograft kidney recipients showing no rejection episodes, no infections and no changes in immuno-suppression therapy in the previous six months. White and grey columns indicate data obtained in healthy controls and kidney transplanted subjects, respectively. (**A**,**B**) Indicate percentage and number of circulating T, CD4^+^ and CD8^+^ T lymphocytes, as indicated; (**C**) Indicates CD4/CD8 ratio; (**D**) Refers CD54 expression level in CD4^+^ and CD8^+^ T lymphocytes, as indicated; as detailed in the Section 4, CD54 expression on the T cell effectors has been expressed as ratio of the mean intensity fluorescence (MIF) value for CD4^+^ and CD8^+^ T cells and the control MIF value obtained after staining the same cell populations with the isotype control mAb, as described [24]. (**E**,**F**) Indicate percentage and number of the circulating T_R3-56_ regulatory T cells, respectively; (**G**,**H**) Refer to percentage and number of the circulating Treg population; (**I**,**J**) Indicate the growth ability of the circulating T_R3-56_ and Treg populations, as represented by their intracellular expression of the ki67 molecule; Statistical evaluation of data was performed by means of the Mann–Whitney test. Statistical significance values are indicated.

**Figure 2 ijms-25-10610-f002:**
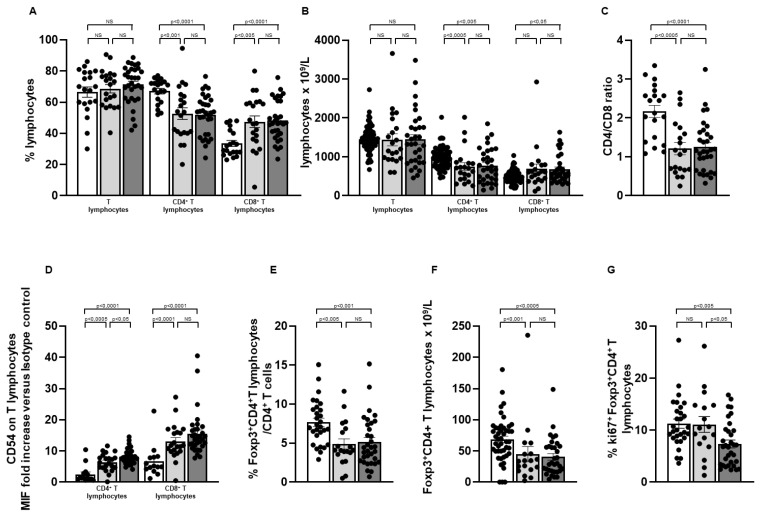
Allograft kidney recipients show association of highest level of circulating T_R3-56_ regulatory T cells with significant decrease in the Treg growth ability and increasing CD54 expression by the CD4^+^ T cell population. White columns indicate healthy controls; light and dark grey columns indicate transplanted subjects showing circulating T_R3-56_ levels <9.16% or ≥9.16% of the T cell population, respectively; the 9.16 cut-off value was obtained by increasing by three standard errors the median value observed in healthy controls (see patient and method section for details). (**A**,**B**) Indicate percentage and number of circulating T, CD4^+^ and CD8^+^ T lymphocytes; (**C**) Shows CD4/CD8 ratio; (**D**) Refers to CD54 expression level in CD4^+^ and CD8^+^ T lymphocytes; as detailed in Section 4, CD54 expression on the T cell effectors was expressed as a ratio of the mean intensity fluorescence (MIF) value for CD4^+^ and CD8^+^ T cells and the control MIF value obtained after staining the same cell populations with the isotype control mAb. (**E**,**F**) Indicate the percentage and number of the circulating Treg in the different cohorts; (**G**) Indicates the growth ability of the circulating Treg population, as represented by their intracellular expression of the ki67 molecule; Statistical evaluation of the data was performed by means of the Mann–Whitney test. Statistical significance values are indicated.

**Figure 3 ijms-25-10610-f003:**
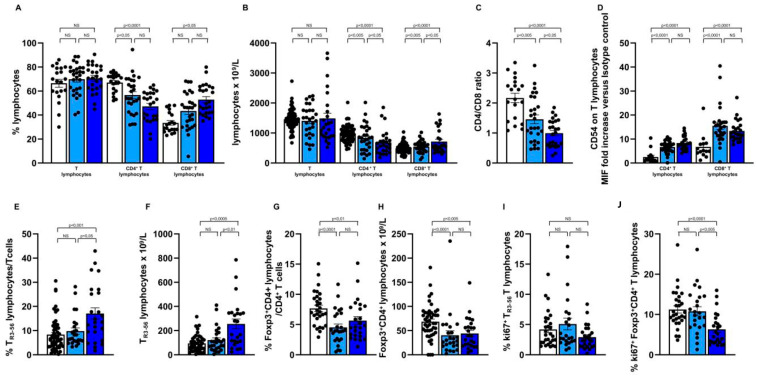
Increasing amount of circulating T_R3-56_ lymphocytes and reduced growth ability of the Treg population characterise kidney transplant recipients showing unstable control of the graft. White columns indicate healthy controls; light and dark blue columns indicate transplanted subjects categorised, according to their clinical and laboratory profile, as belonging to the Stable or Unstable transplant recipient sub-groups, respectively. See Patient and Methods section for details. (**A**,**B**) Indicate percentage and number of circulating T, CD4^+^ and CD8^+^ T lymphocytes; (**C**) Indicates CD4/CD8 ratio; (**D**) Refers to CD54 expression level in CD4^+^ and CD8^+^ T lymphocytes; CD54 expression on the T cell effectors was expressed as a ratio of the mean intensity fluorescence (MIF) value for CD4^+^ and CD8^+^ T cells and the control MIF value obtained after staining the same cell populations with the isotype control mAb, as described [28]. (**E**,**F**) Show percentage and number of the circulating T_R3-56_ lymphocytes; (**G**,**H**) Show percentage and number of circulating Treg; (**I**,**J**) Indicate the growth ability, as represented by their expression of the ki67 molecule, of the circulating T_R3-56_ and Treg population, respectively; Statistical evaluation of data was performed by means of the Mann–Whitney test. Statistical significance values are indicated.

**Table 1 ijms-25-10610-t001:** Higher levels of the circulating T_R3-56_ cells significantly associate with Unstable disease ^1^ condition in a cohort of kidney transplanted subjects showing no infections, no rejection episodes and no changes in immuno-suppression therapy in the previous six months.

	N	Age Mean (Range)	Males/Females	T_R3-56_ /T Cells <9.16 ^2^	T_R3-56_ /T Cells >9.16 ^2^
	50	51.82 (35–68)	31/19	21	29
**Stable Disease ^1^**	25	53.54 (38–68)	15/10	15 ^3^	10 ^4^
**Unstable Disease ^1^**	25	50.32 (35–67)	16/9	6	19

^1^ subgroup categorisation criteria have been detailed in the Patients and Method Section; ^2^ the number has been obtained increasing of three SEM the median value found in the healthy controls for T_R3-56_ percentage/T lymphocytes (see Patient and Method Section for details); ^3^ significant different from the Unstable disease Group (*p* < 0.05 by Fisher exact test; Odd Ratio 0.21 (95% CI: 0.0695 to 0.722); ^4^ significant different from the Unstable disease Group (*p* < 0.05 by Fisher exact test; Odd Ratio 4.75 (95% CI 1.384 to 14.37).

**Table 2 ijms-25-10610-t002:** Characteristics of the subjects enrolled in the study.

Kidney Recipient subjects (N = 53)	
**SEX** M/F (%)	30/23 (56/44)
**AGE** (Mean ± SD)	51.83 ± 14.04
**TRASPLANT VINTAGE** (years)	5.56 ± 4.2

**White Blood Cell count** (×10^9^/L) Mean ± SD	8.278 ± 2.57
**Neutrophil count** (×10^9^/L) Mean ± SD	5.308 ± 2.15
**Lymphocyte count** (×10^9^/L) Mean ± SD	2.029 ± 0.83
**Immunosuppressive drugs**	
**Tacrolimus** Average dosage Mean ± SD	31/536.59 ± 2.63 mg
**Cyclosporine** Average dosage Mean ± SD	22/53168.1 ± 52.18 mg
**Steroids** Average dosage Mean ± SD	53/534.91 ± 2.13 mg
Healthy subjects (N = 20)	
**SEX** M/F; (%)	12/8 (60/40)
**AGE** (Mean ± SD)	45.75 ± 16.28
**White Blood Cell count** (×10^9^/L) Mean ± SD	7.869 ± 1.862
**Neutrophil count** (×10^9^/L) Mean ± SD	5.749 ± 1.378
**Lymphocyte count** (×10^9^/L) Mean ± SD	2.120 ± 0.527

## Data Availability

Data supporting reported results can be obtained from the corresponding author (G.R.).

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
