# Peer review of "TR3-56 and Treg Regulatory T Cell Subsets as Potential Indicators of Graft Tolerance Control in Kidney Transplant Recipients"

_ijms, 2024, doi:10.3390/ijms251910610_

Round 1

Reviewer 1 Report

Comments and Suggestions for Authors

Renal transplantation aims to achieve long-term kidney allograft survival while reducing complications from prolonged immunosuppressive treatment. The researchers studied the immune changes in kidney transplant recipients' cell phenotypes. This manuscript is enjoyable to read. There is a clear rationale for the study, and the results are presented well. Results largely support conclusions.

My comments are:

1. Recent study characterized the variation of cell subtype involved flow cytometry assessment of lymphocyte subpopulations (including Tregs and various lymphocytes) and gene expression analysis of immune-related genes. J Immuno Research 2019; Article 7452019). If they also run PCR, would there be a general difference for gene expression profiling for those patients?

2. a few comments that I hope improve this article.

Title – perhaps the authors would like to highlight the research results

Abstract – requires some editing/rewriting

Introduction - authors may want to shorten the number of paragraphs

Results – the headline information for the subsection is too long

Quality of figures

Fig 1

A,B,D- labeling of the Y-axis is too small

C,E-L, lack of y-axis labeling, hard for the readers to comprehend 

Fig 2

A,B,D- labeling of the Y-axis is too small

C,E-G, lack of y-axis labeling

Fig 3

Same issue as fig 1

Discussion

Authors want to shorten paragraphs.

To have a balanced discussion, authors need to discuss and cite some of the results contradicting their findings. Cheung et al CD4+CD25+ T regulatory cells in renal transplantation. Frontier in Immunology 2022, DOI 10.3389/fimmu.2022.1017683, a nice review, provides ample information for citations.

Comments on the Quality of English Language

minor editing is needed 

Author Response

Reviewer 1 reply

Comment 1 Renal transplantation aims to achieve long-term kidney allograft survival while reducing complications from prolonged immunosuppressive treatment. The researchers studied the immune changes in kidney transplant recipients' cell phenotypes. This manuscript is enjoyable to read. There is a clear rationale for the study, and the results are presented well. Results largely support conclusions.

  1. Recent study characterized the variation of cell subtype involved flow cytometry assessment of lymphocyte subpopulations (including Tregs and various lymphocytes) and gene expression analysis of immune-related genes. J Immuno Research 2019; Article 7452019). If they also run PCR, would there be a general difference for gene expression profiling for those patients?

Reply Thanks for your general comments to the paper. The major aim of our study is to investigate on the immune scenario underlying kidney graft tolerance. With this purpose, in order to identify peculiar immune traits associated with early perturbed control of graft tolerance and to propose new valuable criteria to improve clinical management of kidney transplanted subjects, we enrolled a cohort of graft recipients without signs of kidney rejection, no infectious episodes and no change in the immuno-suppressive regimen in the last 6 months. Subjects with auto-immune diseases as well as with viral infections have been not included.

In this cohort we choose to evaluate the complexity of the immune network underlying graft tolerance induction/maintenance, by using immune-fluorescence and flow cytometry multi-parametric detection, performed in the whole blood. This method, avoiding any purification procedure, allows the direct analysis of the biological complexity of the immune profile in the peripheral blood. Accordingly, we focused on adaptive immune effectors, major responsible for allo-antigen recognition/damaging, by evaluating their surface expression of the CD54 molecule, consistently associated to antigen-dependent T cell activation (Slavin-Chiorini DC et al. Cancer Gene Ther. 2004; Oosten LE et al. Blood 2004). As requested these issues have been focused in the revised paper (lines 520-525)

The relevance of cell-mediated tolerance control has been largely described. Compelling data have been indicating that immune regulatory populations represent a heterogeneous, not completely defined, group of differentiated T cell subsets involved in the prevention of immune-dependent tissue damaging events. In this context, we described that co-expression of CD3 and CD56 molecules identifies the TR3-56 subset as a novel regulatory T cell population specifically involved in the control of cytotoxic T cell function in autoimmunity as well as in multiple haematological contexts (Terrazzano G et al. Nat. Metab. 2020; Carriero F. et al. Cells 2023; Leone S. et al. Eur. J. Haematol. 2022; Rubino V et al Eur. J. Haematol. 2023). Thus, we performed an extended immune profile including such regulatory subset.

As requested, in the discussion section of the revised manuscript, (lines 474-76) we more extensively refer about such key issues, also considering the complementary value of previous reports that, by using gene expression profile (Krajewska et al. 2019) and/or Treg detailed phenotype (Cheung et al. 2022) have been contributing to the analysis/definition of the complex inflammatory pathways underlying kidney transplant microenvironment.

Comment 2. a few comments that I hope improve this article.

Title – perhaps the authors would like to highlight the research results

Abstract – requires some editing/rewriting

Introduction - authors may want to shorten the number of paragraphs

Results – the headline information for the subsection is too long

Quality of figures

Reply As requested, to highlight the main results proposed by the study the title has been modified according with the reviewer suggestions as following: “TR3-56 and Treg regulatory T cell subsets as potential indicator of graft tolerance control in kidney transplant recipients”. (Lines 2-3 of the revised manuscript) Moreover, according with the reviewer request, in the revised manuscript, the introduction section as well as the headline information proposed in the result section have been modified and shortened.

We apologise for the quality of the figures in pdf format; As requested, the labelling of Y axes has been always modified to increase their readability.

Comment 3 Discussion

Authors want to shorten paragraphs.

To have a balanced discussion, authors need to discuss and cite some of the results contradicting their findings. Cheung et al CD4+CD25+ T regulatory cells in renal transplantation. Frontier in Immunology 2022, DOI 10.3389/fimmu.2022.1017683, a nice review, provides ample information for citations.

Reply As requested, in the discussion of the revised manuscript, the paragraphs have been shortened and we have more extensively considered the complementary value of previous reports that, by using gene expression profile (Krajewska et al 2019) and/or Treg detailed phenotype (Cheung et al 2022) have been contributing to the investigation on the complex inflammatory pathways underlying kidney transplant condition. See also point 1 of this reply.

Reviewer 2 Report

Comments and Suggestions for Authors

There are several questions I would like to ask regarding the material and methods section, which is critical to the reliability of your results:

1.  Diabetes mellitus , post-transplant DM and immune dysfunction are interconnected. Did DM or PTDM patient included in this study? if so, how many percentage were they?

2. All the 53 kidney transplant recipients are from cadaveric transplantation. What was the dialysis vintage of these patients? Additionally, patients with CKD and dialysis suffered from chronic immune dysfunction, which may interfere with the presentation of T cells. Could the author explain this confounding factors to their study?

3. What is the proportion of patients who receive thymoglobulin as an induction agent?

4. Could the author mentioned about the average calcineurin inhibitor drugs' level (ex:FK 506)in their cohort?

5. Table 2 regarding the demographic data of studying cohort is too simple. I suggest to add in more information for your cohort, such as issues I raised in my previous questions.

6. Please showed the data of the control group. I recommend the author to modify both tables so that the data of the control group can fix in.

Author Response

Reviewer 2 reply

Comment 1.  Diabetes mellitus, post-transplant DM and immune dysfunction are interconnected. Did DM or PTDM patient included in this study? if so, how many percentage were they?

Reply The major aim of our study is to investigate on the immune scenario, associated with kidney transplant tolerance, to identify/propose peculiar immune traits to be related with early perturbed control of graft tolerance. With this propose we enrolled a cohort of graft recipients without signs of kidney rejection, no infectious episodes and no change in the immuno-suppressive regimen in the last 6 months, regardless the pathogenic condition causing the end-stage kidney disease that required the transplant as key therapeutic option. In addition, as specified in the Patient and Method section, no subjects with auto-immune diseases or viral infections have been enrolled in the study (lines 452-453). As requested, this issue has been specified also in the discussion section of the revised paper (lines 421-424).

  1. All the 53 kidney transplant recipients are from cadaveric transplantation. What was the dialysis vintage of these patients? Additionally, patients with CKD and dialysis suffered from chronic immune dysfunction, which may interfere with the presentation of T cells. Could the author explain this confounding factors to their study?

Thanks for this question. We agree with the reviewer about the ability of the dialysis treatment to affect immune-regulatory network by inducing pro-inflammatory pathways in subjects suffering end-stage kidney failure condition. In order to minimise such possible confounding factor transplant vintage exceeding one year has been considered in the enrolment criteria. In this period, we expect a major role for graft antigenic stimulation as well as for the immune-modulating treatment in shaping the immune scenario of the kidney transplant recipients. As requested, this issue has been adequately discussed in the revised manuscript. (lines 418-424)

  1. What is the proportion of patients who receive thymoglobulin as an induction agent?

No subjects received anti-thymocyte serum in the induction treatment; as requested, this statement has been included in the patient and method section of the revised manuscript (lines 447-448).

  1. Could the author mentioned about the average calcineurin inhibitor drugs' level (ex:FK 506) in their cohort?

As requested, the revised Table 2 refers this issue in a detailed way.

  1. Table 2 regarding the demographic data of studying cohort is too simple. I suggest to add in more information for your cohort, such as issues I raised in my previous questions.

As requested, the revised paper includes a more extensive description of the cohort by us enrolled in the study (see the revised Table 2). See also previous reply points.

  1. Please showed the data of the control group. I recommend the author to modify both tables so that the data of the control group can fix in.

As requested, the modified Table 2, included in the revised manuscript, also shows a more descriptive presentation of the healthy controls enrolled in the study.

Round 2

Reviewer 1 Report

Comments and Suggestions for Authors

The authors intend to address those concerns that I raised for their previous version of the manuscript. This is an important clinical observation as their results suggest there is a causative relationship between the phenotype of Treg cells and the clinical outcome/prognosis of those kidney transplant patients. Nevertheless, as I stated clearly, this is a causative relationship and an important observation. Detailed  mechanistic investigation awaits further work. The authors need to acknowledge this aspect of scientific reality. 

Comments on the Quality of English Language

Moderate language editing is requited. 

Author Response

Comment 1 The authors intend to address those concerns that I raised for their previous version of the manuscript. This is an important clinical observation as their results suggest there is a causative relationship between the phenotype of Treg cells and the clinical outcome/prognosis of those kidney transplant patients. Nevertheless, as I stated clearly, this is a causative relationship and an important observation. Detailed  mechanistic investigation awaits further work. The authors need to acknowledge this aspect of scientific reality. 

Reply 

Thanks again for your general comment on our study. As requested, the observation that the biological mechanisms underlying our results need to be investigated has been added to the revised manuscript (lines 424-25)

Reviewer 2 Report

Comments and Suggestions for Authors

Thank you for taking the time to answer my question. I appreciate the effort you put forth. There are some minor revisions that need to be made before this article can be processed for further consideration:

1. Since the underlying comorbidities of the cohort, including diabetes, chronic kidney disease, and post transplant diabetes, were not categorised in the analysis, I recommend that you add a statement in limitations explaining this.

 2. Table 2 should be reorganized: In order to make comparisons easier for the reader, it would be beneficial to place kidney recipients and healthy subjects side by side. Futhermore, more clinical information is needed to compared these 2 groups: both of their cell counts, their kidney functions, thier nutrition status (parameter such as serum albumin, lipid profiles) and HbA1c.

Author Response

Thank you for taking the time to answer my question. I appreciate the effort you put forth. There are some minor revisions that need to be made before this article can be processed for further consideration:

Comment 1. Since the underlying comorbidities of the cohort, including diabetes, chronic kidney disease, and post transplant diabetes, were not categorised in the analysis, I recommend that you add a statement in limitations explaining this.

Reply. As requested, in the revised version of the paper (Limitations Section) we clearly state the lack of details in the underlying co-morbidities of the enrolled subjects as a limitation of the study (lines 562-64).

Comment 2. Table 2 should be reorganized: In order to make comparisons easier for the reader, it would be beneficial to place kidney recipients and healthy subjects side by side. Futhermore, more clinical information is needed to compared these 2 groups: both of their cell counts, their kidney functions, thier nutrition status (parameter such as serum albumin, lipid profiles) and HbA1c.

Reply. As requested, the Table 2 has been modified showing, side by side, the major clinical features of kidney recipients and healthy subjects; moreover, in order to favour an easier reading of the table,  we preferred to add, in the enrolment criteria, reported in the revised Patient and Method Section of the paper,  also the mean lipid  values of the healthy controls (lines 570-73); the values have been previously indicated only for the graft recipients.
